# Learning Audio-Visual Dynamics Using Scene Graphs for Audio Source Separation

**Moitreya Chatterjee**[1*]
metro.smiles@gmail.com

**Narendra Ahuja**[1]
n-ahuja@illinois.edu

**Anoop Cherian**[2*]
cherian@merl.com

[1]University of Illinois, Urbana-Champaign, Urbana, IL
[2]Mitsubishi Electric Research Labs, Cambridge, MA

## Abstract

There exists an unequivocal distinction between the sound produced by a static source and that produced by a moving one, especially when the source moves towards or away from the microphone. In this paper, we propose to use this connection between audio and visual dynamics for solving two challenging tasks simultaneously, namely: (i) separating audio sources from a mixture using visual cues, and (ii) predicting the 3D visual motion of a sounding source using its separated audio. Towards this end, we present Audio Separator and Motion Predictor (ASMP) – a deep learning framework that leverages the 3D structure of the scene and the motion of sound sources for better audio source separation. At the heart of ASMP is a 2.5D scene graph capturing various objects in the video and their pseudo-3D spatial proximities. This graph is constructed by registering together 2.5D monocular depth predictions from the 2D video frames and associating the 2.5D scene regions with the outputs of an object detector applied on those frames. The ASMP task is then mathematically modeled as the joint problem of: (i) recursively segmenting the 2.5D scene graph into several sub-graphs, each associated with a constituent sound in the input audio mixture (which is then separated) and (ii) predicting the 3D motions of the corresponding sound sources from the separated audio. To empirically evaluate ASMP, we present experiments on two challenging audio-visual datasets, viz. Audio Separation in the Wild (ASIW) and Audio Visual Event (AVE). Our results demonstrate that ASMP achieves a clear improvement in source separation quality, outperforming prior works on both datasets, while also estimating the direction of motion of the sound sources better than other methods.

## 1 Introduction

Events around us are often audio-visual in nature and our senses have evolved to leverage this multimodal synergy to better reason about the world. For example, the sight of a kid laughing aloud while sliding down a slide allows us to associate *the laughing sound* with the kid and get a sense of his/her direction of motion, even when a myriad of other sounds are present in the scene. In this paper, we propose to leverage this synergy between sight and sound for solving two challenging tasks simultaneously, viz.: (i) separating audio sources from a mixture using visual cues, and (ii) the novel task of predicting the 3D visual motion of the sounding source using its corresponding separated audio.

Typical approaches to visually-guided audio source separation use weakly- or self- supervised models trained to separate a mixture of acoustic sources into its constituents by conditioning on appropriate visual regions [8, 13, 55, 56]. Such approaches impose constraints on the space of the separated audio (e.g., cyclic consistency, object identifiability, etc.) to derive gradients for training the underlying

---

*Equal Contribution.

36th Conference on Neural Information Processing Systems (NeurIPS 2022).

neural models. A few of these methods additionally seek to ground the separated audio with the appearance of the sounding object [8, 13, 46]. Chatterjee *et al.* [8], construct a 2D scene graph on the video frames to capture the visual context of the audio source for better separation; their key intuition being that certain sounding objects (e.g., a guitar) cannot produce the sound by itself, but must have a suitable spatial context around them (e.g., a person). However, all the above approaches ignore one key aspect of the physical world – that it is three dimensional and this 3D structure influences the sound being heard. For example, if *a person playing a guitar* is spatially distant from the microphone, then the audio mixture to be separated is unlikely to have the sound of that guitar. Further, the 3D scene structure also allows for incorporating the motion of sounding objects. Imagine *the whistling of a train moving towards you*. There is an inevitable relationship between the evolution of the sound of the train being heard and its 3D motion, which can help distinctly separate its sound from other whistling trains or background clamour and vice-versa. Thus, if the audio of this train is well-separated via visual-guidance, then the separated audio must also be able to predict the direction of the train's motion. Leveraging these insights, we present Audio Separator and Motion Predictor (ASMP) – an innovative graph neural network for video-guided audio source separation from an acoustic mixture that can also predict the direction of motion of the sound source.

Inspired by Cherian et al. [9], our ASMP framework begins by computing a dense 2.5D representation of the frames of a video where 2.5D refers to the 2D visual context of the frames enriched with the pseudo-depth for that frame produced using a 2D-to-3D monocular depth prediction method [39]. Next, we succinctly capture the semantic context of this 2.5D visual scene by means of a novel 2.5D scene graph representation. The nodes of this scene graph capture the various objects in the scene (projected onto the pseudo-3D scene via the depth map), while the graph edges characterize the approximate 3D spatial distances between the objects. Note that our graph does not explicitly capture any spatio-temporal dynamics of the scene objects, instead is constructed on singleton frames. To achieve audio source separation, we propose a recurrent graph neural network that is trained to segment this 2.5D scene graph into sub-graph embeddings; each of which is trained to be associated with a potentially unique *sounding object or interaction* and is used to induce separation of that sound source from the audio mixture. During training, we enforce this uniqueness via imposing orthogonality constraints between the generated sub-graph embeddings. To make ASMP associate the evolution of sound with the 3D motions of their sources, we propose to include an auxiliary task that demands the prediction of the 3D direction of motion of a sounding object from its separated audio, where the ground truth 3D motion is estimated from the 2.5D scene graph using optical flow.

A natural question one may have at this point is: *how can a method predict the 3D motion from monaural audio using a 2.5D scene graph constructed on a single video frame?* The key insights come from two observations: (i) there are often implicit object motion cues present in singleton video frames, which may be embedded within the scene graph node features (e.g., a graph node corresponding to a *train* may have the inductive priors suggesting the train is moving in the direction it faces), and (ii) when the audio is separated via visual-guidance, these implicit features modulate the audio separation masks, thereby incorporating motion cues into the separated audio spectrograms. Thus, when trained end-to-end for the joint task of sound separation and motion prediction, the model leverages these implicit cues to minimize the learning objective.

To demonstrate the efficacy of ASMP on the dual tasks of sound source separation and motion direction prediction, we present experiments on two challenging datasets, namely (i) Audio Separation in the Wild (ASIW) [8] and (ii) Audio Visual Event (AVE) [47]. Both of these datasets feature videos of sounding objects and sounding interactions *in the wild*. Our results clearly show that ASMP outperforms competing prior methods on both these datasets as well as on both the tasks, underscoring the importance of incorporating 3D spatial structure and the benefits of learning audio-visual dynamics.

Below, we summarize the key contributions of the paper:

- We introduce a novel *3D geometry-aware scene graph representation* [20] for visually-guided audio source separation, called ASMP.
- We introduce a novel task of *predicting 3D motion direction* of a sound source in the scene from the the temporal evolution of the sound it makes, aided by appropriate visual context, and use it to improve audio separation.
- ASMP demonstrates *state-of-the-art* audio source separation and motion prediction performances on two challenging datasets for this task, viz. ASIW and AVE.

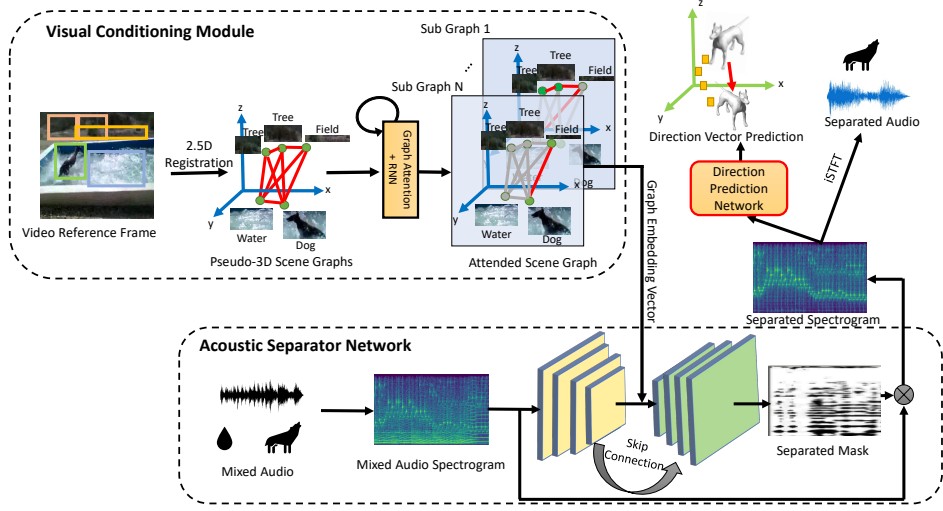

Figure 1: A detailed illustration of our proposed ASMP model.

## 2 Related Works

Below, we briefly review some of the important prior works closely related to our approach.

**Visually-guided Audio Source Separation** is the task of separating a mixture of audio signals into its constituent sources by discovering the association between the separated sound sources and their visual appearances in the physical world. Typical methods for solving this task find a wide range of real-world applications, such as separating sounds of musical instruments [13, 8, 11, 55, 56], separating speech signals [1, 10, 33], and separating on-screen sounds from off-screen ones for generic objects [36, 48]. A learning pipeline often used to train such methods conditions a sound-separation network (such as a U-net [41]) with an appropriate visual object representation, to induce a source separation on a mixed audio input, obtained by mixing the audio streams of two different videos. In recent years, research in this area has moved from using global visual features of motion and appearance [55] towards extracting very fine-grained visual conditioning information to induce more effective separations [13, 46]. Additional refinement steps have also been explored [8] to ward off potential false separation triggers. However, none of these approaches factor in the 3D geometry of the scene captured by the video – an important gap that we attempt to fill using ASMP.

**Localizing Sound in Video Frames** has been attempted in several recent methods by learning to ground the sound source in the visual space, e.g., identifying the pixels of the sound source [4, 17, 21, 42, 46]. While, these approaches strive to learn the association between the visual appearance and acoustic signatures of sound sources, they do not apply such methods to audio source separation – a task that is the focus of this work.

**Sound Synthesis from Videos** constitutes another category of important techniques in the audio-visual realm [37, 57] that has become popular lately. Towards this end, several approaches have recently been proposed for the task of generating both monaural and binaural audio starting from videos [12, 34, 53, 27]. However, differently, we seek to solve the task of sound source separation and motion prediction in the visual space.

**Application of Scene Graphs to Videos** has resulted in massive strides in video understanding tasks. Scene graphs, while traditionally used for capturing the static content of images [20, 28] have lately been used for several video understanding tasks. For instance, Ji *et al.* [19] applied them for action recognition, Geng *et al.* [14] for visual dialog, and Chatterjee *et al.* [8] for visually-guided sound-source separation. However, these scene graphs are usually 2D, while we attempt to explicitly incorporate the 3D scene geometry into the scene graphs. While, Cherian et al. [9] proposes (2.5+1)D scene graphs for video question answering, our task of audio source separation and motion prediction brings in several novel components beyond their setup.

**Audio Separation Using Direction of Arrival** has been an important topic of recent interest in the audio research community [52, 31, 35, 44], where the direction of arrival of sound to a microphone array is explicitly used for improved sound source separation. While, our approach is inspired by their key findings, we attempt to explore this in the audio-visual domain.

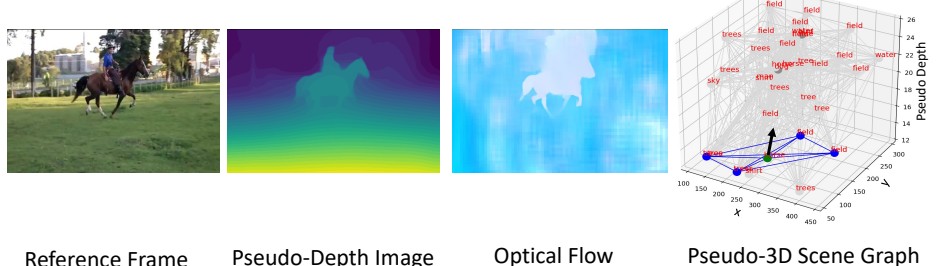

| Reference Frame | Pseudo-Depth Image | Optical Flow | Pseudo-3D Scene Graph |

Figure 2: A step-wise illustration of the pseudo-3D scene graph construction process on an example frame from an ASIW video. We show the original (reference) frame, its predicted monocular 3D depth map, its optical flow estimate with an adjacent frame, and a pseudo-3D scene graph for the frame. The highlighted part of the graph (blue edges) is the sub-graph associated with the sound of the horse's motion, while the black-arrow denotes the motion direction of the horse in pseudo-3D space. Note that the motion direction is not used in our graph embeddings, instead provides ground truth guidance when minimizing our losses.

## 3 Proposed Method

### 3.1 Task Setup and Method Overview

Let $V$ denote an unlabeled video, and $m(t) = \sum_{i=1}^{N} \mathbf{a}_i(t)$ be its accompanying discrete-time audio arising from a linear mixture of $N$ acoustic sources $\mathbf{a}_i(t)$. The main goal of *visually-guided audio source separation* is to use cues derived from $V$ to induce a separation of $m(t)$ into its constituent acoustic sources $\mathbf{a}_i(t)$, for $i \in \{1, 2, \ldots, N\}$. For effectively capturing the audio-visual semantic alignment, we represent the video $V$ as a scene graph $\mathcal{G} = (\mathcal{V}, \mathcal{E})$ with nodes $\mathcal{V} = \{n_1, n_2, \ldots, n_K\}$ denoting the objects in $V$ and $\mathcal{E}$ capturing the set of edges $e_{ij}$ characterizing the spatial proximity of a pair of nodes $(n_i, n_j)$. The key idea in our proposed Audio Separator and Motion Predictor (ASMP) framework is to map each of the acoustic sources $\mathbf{a}_i(t)$ to a sub-graph of $\mathcal{G}$. We realize this mapping by auto-regressively partitioning the graph $\mathcal{G}$ into mutually-orthogonal neural embeddings and using these embeddings to condition an *Acoustic Separator* sub-network tasked with extracting a sound source $\mathbf{a}_i(t)$ from the mixture $m(t)$. A key ingredient in our model is equipping the audio separator to also predict the direction of motion of the sounding object. We incorporate this ability by including a direction prediction training loss. Figure 1 provides an overview of our model and the details follow.

### 3.2 Estimating Ground Truth Sound Source Motion

One novel aspect of ASMP is that it can predict the motion direction of a source from its sound, which in turn is derived from an embedding of a sub-graph of $\mathcal{G}$. However, the ground truth for this task is not directly available from the 2D frames of a given video. In the following, we detail the pre-processing steps for estimating this ground truth, and present an overview of these steps in Figure 2. We denote a motion displacement vector, as $\mathbf{d}_i$ for the source $\mathbf{a}_i(t)$, which will be subsequently used as an auxiliary cue to train the audio source separator.

**Computing Pseudo-Depth Maps**: Constructing a 3D scene from 2D images is a classic problem explored in computer vision for which a variety of solutions exist [2, 15]. However, many of these methods make strong assumptions, such as: (i) the scene being static, (ii) existence of sufficient overlap between the frames, or (iii) knowledge of the intrinsic parameters of the camera, none of which might hold for the type of videos that are typically used in our task. To this end, we propose to build upon the recent progress made in monocular 3D reconstruction realm towards mapping each video frame $F_f$ of $V$ into a 2.5D pseudo-depth image [54]. In particular, we use the MiDAS algorithm [39] because of its ease-of-use and accuracy of depth predictions.

**Rectifying Frames for Camera Motion:** Since we do not assume that the frames in $V$ have been captured by a static camera, we next align them to a common 3D reference frame $F_r$. To this end, we apply the classic *Lucas-Kanade* tracker [30] across the frames. As there may be objects moving out of the scene or shot-switches in the video, we found that tracking the *static objects* in the video typically does not sustain its full length. Thus, we group the frames into fixed-size windows, each with a maximum of $\ell$ continuous frames. A video can therefore, be considered as a collection of $W$ windows, with each window producing its own motion displacement vector. This consequently

also implies that our frames are aligned (rectified) within their respective windows. Concretely, let $\mathbf{x}$ denote a tracked point, in pseudo 3D, within a window. Suppose $\mathbf{x}_f \in \mathbb{R}^3$ and $\mathbf{x}_r \in \mathbb{R}^3$ be the 3D coordinate corresponding to $\mathbf{x}$ in $F_f$ and that in the reference frame $F_r$, respectively. Then, using the full set of such tracked points in this window, we rectify $F_f$ to $F_r$ by estimating a 3D transformation matrix, accurate upto a scale, using the iterative closest point (ICP) algorithm [6]. Once the target frame $F_f$ has been rectified, we also rectify its corresponding pseudo-depth map using the same transformation matrix.

**Visual Detection of Sound Sources:** To construct our scene graph, we first use a Faster-RCNN (FRCNN) [40] model for detecting a set of $M$ objects and their 2D spatial bounding boxes in the reference frame $F_r$ of a window. The detection model is trained (on the Visual Genome dataset [25]) to recognize and localize 1600 commonplace object classes. However, some of the data that we use in our experimental study consists of sound producing objects that are not in the Visual Genome classes list. To accommodate such classes (e.g., musical instruments), we use a separate FRCNN model pre-trained on images annotated with these classes in the Open Images dataset [24]. Let $\mathcal{C}$ denote the set of all detectable classes by either of these two detectors. For a frame $F_r$, the FRCNN model produces a set of $M$ quadruples: $\left\{ \left( C_r^i, B_r^i, F_r^i, S_r^i \right) \right\}_{i=1}^M = \text{FRCNN}(F_r)$, one for each detected object, consisting of the label $C \in \mathcal{C}$ of the detected object, its bounding box $B$ in the frame, a feature vector $F$ characterizing the visual appearance of the object, and a detection confidence $S$.

**Tracking a Sound Source:** In order to robustly track the visual trail of a potential sound source between a reference frame $F_r$ and a target frame $F_f$, we employ RAFT [45] – an off-the-shelf state-of-the-art robust optical flow method. RAFT yields a dense optical flow field for each pixel in the reference frame. We filter this flow field using the bounding boxes $B_r^i$, and associate it with the $i$-th detected sounding object. Next, we base the flow field in a 3D coordinate space using the estimated pseudo-depth map associated with the respective frames; thus allowing for the recovery of the motion track of the sounding object in a pseudo-3D space.

**Estimating the Ground-Truth Displacement Vector:** Our proposed flow-based tracking method yields numerous displacement vectors (one for each pixel) within a detected object bounding box, this box corresponding to a potential sound source in the reference frame $F_r$. However, as alluded to above, we envisage representing a scene succinctly using a scene graph with each node capturing a single motion vector. To this end, we summarize the flow field associated with a detection box using its median; thus charaterizing the 3D displacement $\mathbf{d}_i$ of that object in that window.

### 3.3 Constructing the Pseudo-3D Visual Scene Graph

With the object detections in place, as described above, we now proceed to construct the *Pseudo-3D Visual Scene Graph* for $V$. While, for most of the prior approaches [14, 19], all detected objects are trivially added to the graph as nodes, our use-case of sound source separation dictates that the constructed graph permits the learning of correlations between appropriate subgraphs and the separated audio sources. Towards this end, each sound source in a dataset is associated with a set of objects in the visual domain (i.e., from the set of classes $\mathcal{C}$ of FRCNN) which we call *Auditory Objects* and which could potentially create sound. For instance, a *crying* sound could arise from classes such as *kid*, *baby*, etc. Let us denote an auditory object by $p$, where $p \in \mathcal{C}$. The entire video $V$ could then be thought of as a collection of auditory objects, $P = \{p_1, \ldots, p_N\}$, which together generate the mixed audio $m(t)$. For each of these auditory objects $p_i$, we identify the subset $\mathcal{V}_{p_i}$ of $M - N$ objects that are non-auditory and overlap with $p_i$ with an Intersection Over Union (IoU) greater than a pre-defined threshold $\gamma$. These objects are called *Context Objects*. The vertex set $\mathcal{V}$ of the graph $\mathcal{G}$ associated with the video $V$ is then constructed as $\mathcal{V} = \bigcup_{i=1}^N (\{p_i\} \bigcup \mathcal{V}_{p_i})$.

Next, we describe the process of constructing the edge set $\mathcal{E}$ of the graph $\mathcal{G}$. While prior approaches that use scene-graphs, typically either design dense fully-connected scene-graphs with equal edge weights [8] or deploy a visual relationship detector [14], nonetheless they do not usually embody any 3D scene geometry. Instead in ASMP, we propose to mitigate this shortcoming by computing the edge weights from the pseudo-3D scene we constructed above. To this end, we first extract 3D point clouds for every object $i$ in frame $F_r$ by leveraging the rectified pseudo-depth map. Let $Q_i$ (for $i = 1, 2, \ldots M$) denote the pseudo 3D point cloud of all pixels associated with node $n_i$ within its corresponding FRCNN bounding box. Note that each point $\mathbf{x}_r^{Q_i} \in Q_i$ is a 3D vector with pixel-normalized $\mathbf{xy}$ coordinates and their pseudo-depths. To define the spatial proximity between two nodes $n_i$ and $n_j$, we compute the symmetrized Chamfer distance ($D_{ij}$) [5] between every pair

of points in $\mathbf{x}_r^{Q_i}$ and $\mathbf{x}_r^{Q_j}$. These distances are then normalized to be within $[0, 1]$ by dividing by the maximum value over all edges in $\mathcal{E}$ and are used to create a weighted-adjacency matrix defining the graph edges $\mathcal{E}$ by using a radial basis function on the Chamfer distance matrix, i.e., the weight of an edge $e_{ij} = \exp(\frac{-D_{ij}}{\sigma^2})$, for a suitable scale $\sigma$, set in our experiments as the $25^{th}$ percentile of the entries in the distance matrix.

**Visual Encoding of Sounding Interactions:** The scene graph $\mathcal{G}$ is a spatio-semantic summary of the visual information in $V$ and is the visual analogue to the mixed audio $m(t)$ accompanying $V$. For the purposes of our task, we are interested in extracting appropriate subgraphs of $\mathcal{G}$ that are capable of inducing a separation of $m(t)$ into its constituents $\mathbf{a}_i(t)$ for $i = \{1, 2, \ldots, N\}$. This is accomplished by employing a recurrent graph attention to segment the graph $\mathcal{G}$ into sub-graphs. To segment the graph $\mathcal{G}$, we first employ a multi-headed graph attention network [49] which operates on the features associated with the scene graph nodes and performs multi-head graph message passing. This allows ASMP to incorporate the spatial scene geometry (from edge weights) and scene context from the neighbors of an auditory node for characterizing the visual representation of a sound source. For capturing the visual representation of interactions (between objects) that produce a sound, we design an edge convolution network [51], the features from which are then simultaneously pooled with the graph attention features using global max-pooling and global average pooling [26] and concatenated. This results in an embedding vector $\boldsymbol{\zeta}$ for the entire graph. Since we need to extract separate sub-graph embeddings to induce a separation of the sound mixture into $N$ sources, we employ a gated recurrent unit (GRU), to recursively extract visual feature vectors $Y = \{\boldsymbol{y}_1, \boldsymbol{y}_2, \ldots, \boldsymbol{y}_N\}$ from $\boldsymbol{\zeta}$. We noticed that extracting a separate background sound from every mixture helps. Thus, our GRU is executed for $N + 1$ steps, rather than $N$.

To ensure that the extracted graph embeddings $Y$ from the GRU do not repeat a sub-graph, we enforce mutual orthogonality [8] between these embeddings. That is, each embedding $\boldsymbol{y}_i$ emerging out of a recurrence of the GRU is expected to produce a unit-normalized embedding $\hat{\boldsymbol{y}}_i$ that is orthogonal to each of the unit-normalized embeddings generated prior to it, i.e., $\{\hat{\boldsymbol{y}}_1, \hat{\boldsymbol{y}}_2, \ldots, \hat{\boldsymbol{y}}_{i-1}\}$. This constraint is incorporated as a regularization in our training setup. Mathematically, a softer-version of this constraint given by the following is implemented:

$$\mathcal{L}_{\text{ortho}}(Y) = \sum_{i,j \in \{1,2,\ldots,N\}, i \neq j} (\hat{\boldsymbol{y}}_i^\top \hat{\boldsymbol{y}}_j)^2. \tag{1}$$

**Acoustic Separator Network:** Finally, to induce the separation in the acoustic signal, we incorporate an *Acoustic Separator Network* (ASN), which is a U-Net [41] style encoder-decoder network. Such networks have recently shown promise in sound source separation tasks [18, 29], particularly those that are employed in conditioned settings [13, 32, 43, 56]. The network has three main parts: (i) an *encoder* consisting of a stack of 2D-convolution layers, each coupled with Batch Normalization and Leaky ReLU, (ii) a *bottleneck* layer that concatenates the encoder embedding with the conditioning information, and (iii) a *decoder* which is made up of a series of up-convolution layers, followed by non-linear activations, each coupled with a skip connection from a corresponding layer in the U-Net encoder, matching in spatial resolution of its output. The encoder encodes the input, which is a magnitude spectrogram $\mathbf{X} \in \mathbb{R}^{\Omega \times T}$ of a mixed audio $m(t)$ produced via the short-time Fourier transform (STFT), where $\Omega$ and $T$ denote the number of frequency bins and the number of video frames, respectively. This encoding is then concatenated with each of the normalized visual encoding vectors $\hat{\boldsymbol{y}}_i$ and decoded to generate a time-frequency mask, $\hat{\mathbf{M}}_i \in [0, 1]^{\Omega \times T}$, which when multiplied with the magnitude spectrogram $\mathbf{X}$ of the mixture yields an estimate of the magnitude spectrogram of the separated source $\hat{\mathbf{S}}_i = \hat{\mathbf{M}}_i \odot \mathbf{X}$, where $\odot$ denotes element-wise product. The separated signal in time domain, $\hat{\mathbf{a}}_i(t)$, is then recoverable by applying inverse Short time Fourier Transform (iSTFT).

**Direction Prediction Network:** A single-channel audio typically only indicates whether the sound source is moving towards/farther away from the camera [3]. However the sub-graph embeddings, $\boldsymbol{y}_i$, that the ASN is conditioned on, injects into the separated spectrograms, motion information implicit in the appearance of the objects in the frame. This enriches these spectrograms, permitting us to obtain a coarse estimate of the direction of source motion in pseudo 3D. We therefore cast the task of predicting the motion vector $\mathbf{d}_i$ of the auditory object as a classification task. We setup two variants of this task using varied quantizations of the unit displacement vector $\mathbf{d}_i$, namely: (i) to predict the octant (or corners) of a unit-cube into which an 8-quantized unit displacement vector $\mathbf{d}_i$ belongs, where the center of the cube refers to the 3D centroid of the corresponding graph node; and (ii) to predict among the 26 directions pertaining to the 8 corners, 6 faces, and 12 edges of a unit-cube, $\mathbf{d}_i$ is

quantized into (using cosine similarity). Additionally, for both settings a class is created for denoting little/no motion and another class captures the direction vector obtained from the background source. We thus have 10 and 28-classes respectively, in the two settings. We realize this objective in ASMP by incorporating a classifier network $h_\Lambda(\cdot)$, with parameters $\Lambda$, that takes as input the (spectrogram) output of the ASN and produces the class label corresponding to the quantization of $\mathbf{d}_i$.

Note that we compute a separate displacement vector $\mathbf{d}_i$ per window $w$, while the separated $\hat{\mathbf{S}}_i$ spans the whole length of the video. We thus compute a set of displacement vectors $\{\mathbf{d}_i^w\}_{w=1}^W$, for each window $w$. Our network takes as input the separated spectrogram, $\hat{\mathbf{S}}_i$, and the visual encoding of the source $i$ ($\hat{\boldsymbol{y}}_i$) and produces as output an estimate of the direction class, $\hat{\mathbf{d}}_i^w \in \{0, 1, 2, \ldots, 9\}$ or $\hat{\mathbf{d}}_i^w \in \{0, 1, 2, \ldots, 27\}$, depending on the direction in which the vector lies. In order to predict the vector for a particular window, appropriate time-aligned slices of $\hat{\mathbf{S}}_i$ are chosen. The sliced spectrogram is then embedded through a ResNet [16] style network. Thus, we have $h_\Lambda : \hat{\mathbf{S}}_i \to \left\{\hat{\mathbf{d}}_i^w\right\}_{w=1}^W$, where the prediction for each window is treated as a different sample in the batch. Details in the appendix.

### 3.4 Learning Losses

ASMP trains purely with weak/self-supervisory cues. In particular, we train our model by employing the "mix-and-separate" strategy [8, 11, 13, 55, 56], where the audio tracks from multiple videos (typically two) are combined, with the goal being to separate the audio for each possible source from the mixture ($\mathbf{X}$). Towards this end, we employ the following set of losses:

**Consistency Loss** ensures that the separated spectrograms of a particular class of sound source (such as a sound from a guitar, the cry of a baby, etc.) always retains its (visual) identity irrespective of which video it comes from [13]. Typically a cross-entropy loss satisfies this requirement. However in ASMP, since the separation is induced by a holistic scene graph recurrently embedded by a GRU, it is not known in advance, in what order the sources are separated. Thus, we consider all possible combinations of the ground truth labels and assign the one with the minimum loss. This is given by:

$$\mathcal{L}_{\text{cons}} = \sum_{u=1,2} \min_{\sigma^u \in \mathcal{S}_{N_u+1}} -\sum_{i=1}^{N_u+1}\sum_{c=1}^{K} \mathbb{1}_{i,\sigma^u(c)}^u \log p_{i,c}^u, \tag{2}$$

where $N_u + 1$ is the number of auditory objects in the $u$-th video, $\mathcal{S}_{N_u+1}$ indicates the set of all permutations on $\{1, \ldots, N_u + 1\}$, $p_{i,c}^u$ denotes the predicted probability produced by the classifier for class $c$ given $\hat{\mathbf{S}}_i^u$ as input, $\mathbb{1}_{i,\sigma^u(c)}^u$ is an indicator for the ground-truth class of the $c$-th object in video $u$, and $K$ denotes the number of sounding object classes in the dataset.

**Cyclic Loss** attempts to preserve the consistency of the generated audio masks from the same video. Specifically, this loss ensures that the separated masks corresponding to each of the sources in the video, when combined, yields a mask which separates the audio spectrogram for that video from the mixture audio (that combines audio from two videos). Mathematically, our cyclic loss is written as: $\mathcal{L}_{\text{cyc}} = \sum_{u=1,2} \left\| \sum_{i=1}^{N_u+1} \hat{\mathbf{M}}_i^u - \mathbf{M}_{\text{ibm}}^{\mathbf{u}} \right\|_1$, where $\mathbf{M}_{\text{ibm}}^{\mathbf{u}} = \mathbb{1}_{\mathbf{X}^u > \mathbf{X}^{\neg u}}$ denotes the binary mask for the audio of video $u$ within the mixture $\mathbf{X}$.

**Direction Prediction Loss** is a standard cross-entropy loss, which checks for the accuracy of the predicted direction class $\{0, 1, \ldots, D_k\}$, where $D_k \in 9, 27$ for every separated sound source, per window. It is given by:

$$\mathcal{L}_{\text{dirpred}} = \sum_{u=1,2}\sum_{w=1}^W \min_{\sigma^u \in \mathcal{S}_{N_u+1}} -\sum_{i=1}^{N_u+1}\sum_{c=1}^{D_k} \mathbb{1}_{i,\sigma^u(c)}^{u,w} \log q_{i,c}^{u,w}, \tag{3}$$

where $N_u + 1$ is the number of auditory objects in the $u^{th}$ video. $\mathcal{S}_{N_u+1}$ indicates the set of all permutations on $\{1, \ldots, N_u + 1\}$, $q_{i,c}^{u,w}$ denotes the predicted probability produced by the classifier for class $c$ given $\hat{\mathbf{S}}_i^{u,w}$ as input, and $\mathbb{1}_{i,\sigma^u(c)}^{u,w}$ is an indicator of the ground-truth class of the $c$-th object in video $u$ for window $w$.

**Final Training Loss** of ASMP is given by the following, with weights $\lambda_1, \lambda_2, \lambda_3, \lambda_4 > 0$:
$$\mathcal{L} = \lambda_1\mathcal{L}_{\text{cons}} + \lambda_2\mathcal{L}_{\text{cyc}} + \lambda_3\mathcal{L}_{\text{ortho}} + \lambda_4\mathcal{L}_{\text{dirpred}}. \tag{4}$$

Table 1: SDR, SIR, and SAR results on the ASIW and AVE test sets. [Key: **Best**, second-best results.]

| Approach | ASIW | | | AVE | | |
|---|---|---|---|---|---|---|
| | SDR ↑ | SIR ↑ | SAR ↑ | SDR ↑ | SIR ↑ | SAR ↑ |
| Sound of Motion (SofM) [55] | 6.7 | 9.4 | 11.1 | 4.1 | 9.2 | 7.6 |
| Cyclic Co-Learn [46] | 7.0 | 13.4 | 12.4 | 4.2 | 9.7 | 8.4 |
| Co-Separation [13] | 6.6 | 12.9 | 12.6 | 3.9 | 9.3 | 7.8 |
| AVSGS [8] | 8.8 | 14.1 | 13.0 | 5.8 | 10.4 | 8.2 |
| ASMP (only 2.5D graph) | 9.0 | 14.3 | 13.7 | 6.5 | 12.4 | 8.9 |
| ASMP (2.5D graph + motion) | **9.6** | **14.5** | **14.1** | **7.2** | **13.3** | **9.4** |

Table 2: Direction Prediction results on the ASIW and AVE on test splits.

| Direction Prediction | ASIW | | AVE | |
|---|---|---|---|---|
| | 10-class (%)↑ | 28-class (%)↑ | 10-class (%)↑ | 28-class (%)↑ |
| Majority Vote | 27.3 | 25.4 | 29.2 | 24.3 |
| Sound of Motion (SofM) [55] | 29.6 | 27.0 | 31.2 | 30.6 |
| Cyclic Co-Learn [46] | 34.8 | 32.3 | 30.7 | 29.2 |
| Co-Separation [13] | 32.2 | 31.7 | 30.2 | 28.0 |
| AVSGS [8] | 39.2 | 38.7 | **38.9** | 34.7 |
| ASMP (Ours) | **42.5** | **41.3** | 38.5 | **36.8** |

## 4 Experiments

In this section, we demonstrate the effectiveness of our ASMP approach on challenging visually-guided audio separation benchmarks.

### 4.1 Datasets and Experimental Setup

**Audio Separation in the Wild (ASIW) Dataset:** Several prior works present experiments mainly for separating sounds of musical instruments [11, 46]. Such results may not necessarily carry over easily to more generic and natural day-to-day life contexts. To this end, we consider the recently proposed *Audio Separation in the Wild* (ASIW) dataset [8] for our experiments. This dataset consists of 147 validation, 322 test, and 10,540 training videos crawled from the larger AudioCaps dataset [22]. It has 14 auditory classes (i.e. $K = 15$, 14 auditory classes and 1 background audio class), such as *baby cries*, *bell ring*, *birds chirp*, *camera clicks*, *train sounds*, *automobile sounds*, etc. Each video in the dataset is 10s long, has significant camera motion, and captures diverse audio-visual contexts. To evaluate ASMP for direction prediction, we divide each video into temporal windows of 1s long and predict a discretized 3D direction vector for a randomly chosen window.

**Audio Visual Event (AVE) Dataset**: Apart from the challenging ASIW dataset, we conduct additional experiments by adapting the popular *Audio Visual Event* (AVE) Dataset [47] for our task. Since this dataset was originally designed for the task of identifying audio-visual events, adapting it to evaluate our approach is straightforward. We treat each audio-visual event class as an auditory class and associate with it a set of *Auditory Objects* (as discussed before). The dataset contains 2211 training, 257 validation, and 261 test set videos. We use videos corresponding to 18 audio-visual event classes (i.e. $K = 19$ here), including both potentially-moving object classes such, such as bus, train, etc. and static classes like banjo, clock etc. The videos in this dataset are 10s long as well. Similar to ASIW, we divide each video into windows of 1s long for computing the motion displacement vectors.

**Baselines:** To the best of our knowledge, the task of jointly separating audio sources (using video) and predicting motion directions is a novel task, and thus does not have a prior baseline to compare to. Therefore, we compare ASMP against the following state-of-the-art methods for audio-visual source separation, specifically those methods that use the "mix-and-separate" learning and also evaluate against them for assessing the efficacy of direction prediction. Specifically, we compare to: (i) *Co-Separation* [13] that uses a single auditory object for conditioning a source separation network,(ii) *Sound of Motion* (SofM) [55] that incorporates object/human appearances and their pixel-level motion

Table 3: SDR, SIR, and SAR Ablation results on the ASIW test set. [Best results in **bold**.]

| Row | Method | ASIW | | |
|---|---|---|---|---|
| | | SDR [dB]↑ | SIR [dB]↑ | SAR [dB]↑ |
| 1 | ASMP (Full Model) | **9.6** | **14.5** | **14.1** |
| 2 | ASMP- Multiscale Chamfer | 9.2 | 14.1 | 14.0 |
| 3 | ASMP- Only 10-class Direction Prediction ($\lambda_1 = \lambda_2 = \lambda_3 = 0$) | 6.4 | 11.2 | 11.7 |

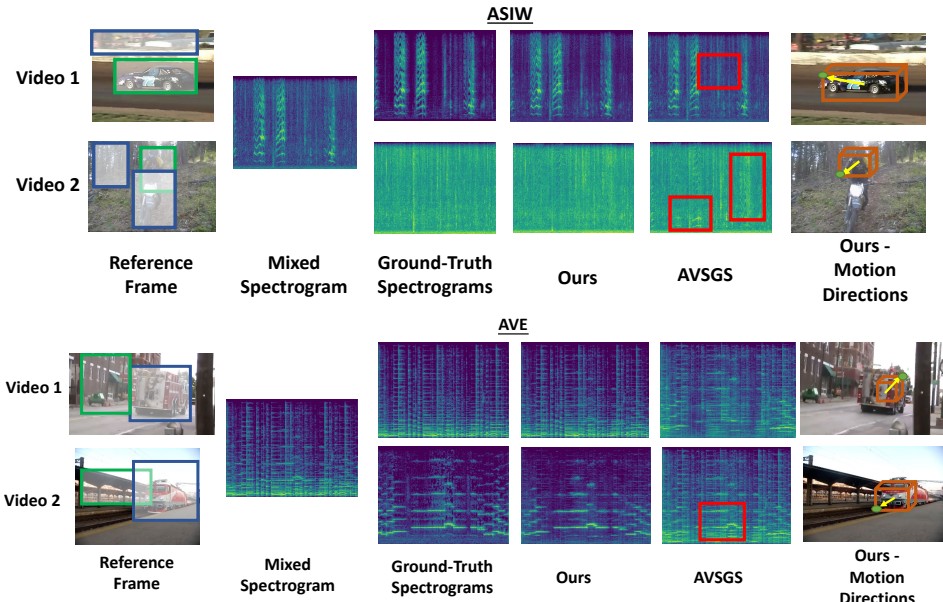

Figure 3: Audio source separation and motion prediction results on ASIW (top) and AVE (bottom). **Bounding boxes** on reference frames show regions attended by ASMP. Red boxes indicate regions of high differences between ground truth and predicted spectrograms. The orange unit cube shown in the *Motion Directions* images, indicates the object whose motion is predicted. The green dot shows the ground truth motion direction, while the yellow arrow shows the predicted direction by ASMP.

trajectories to condition an audio separation network, (iii) *Cyclic Co-Learning* [46], which is a recent method that trains a deep neural model for the joint task of audio separation and visual grounding of the audio source, and (iv) *Audio-Visual Scene Graph Segmenter* (AVSGS) [8], that uses a scene graph for conditioning an audio separator network using the 2D visual context of the audio source.

**Evaluation Metrics:** We quantify the audio source separation performances using the standard evaluation metrics [13, 46, 56], namely: (i) *Signal-to-Distortion Ratio (SDR)* (in dB) [38, 50] – a higher SDR indicating a more faithful reproduction of the original signal, (ii) *Signal-to-Interference Ratio (SIR)* (in dB) – quantifying the extent of reduction in interference in the estimated signal, and (iii) *Signal-to-Artifact Ratio (SAR)* (in dB) – capturing the extent to which artifacts are introduced by the separator. Further, as our motion direction prediction sub-task is cast as a classification problem in our setup, we also report its classification accuracy (Dir. Acc. (%)) for both 10 and 28 class settings.

**Implementation Details:** For every video in our setup, we detect upto two auditory objects, one background auditory object, and 20 context nodes. The visual feature of the background object is derived from a FRCNN embedding of a random crop of the reference frame $F_r$, from a region which does not overlap with other boxes in the frame. The audio streams are sub-sampled at 11kHz and STFT spectrograms are extracted using a Hann window of size 1022, with hop length of 256, following prior works [13, 56]. We set $\Omega = 256$ and $T = 256$. The embeddings, $\hat{y}_i$, and the GRU-hidden state are 512-dimensional. The IoU threshold, $\gamma$ is set to 0.1 for both datasets. Each window in a video has $l = 8$ frames. The weights on the different losses are as follows: $\lambda_1 = 0.05, \lambda_2 = 1.0, \lambda_3 = 1.0, \lambda_4 = 0.05$. Our model is trained using the ADAM optimizer [23] with a weight decay of 1e-4, $\beta_1 = 0.9, \beta_2 = 0.999$. The learning rate is set to $1e-4$ and is decreased by a factor of 0.1 every 15K iterations.

## 4.2 Experimental Results

In Table 1, we present results for source separation by our method versus competing approaches on the ASIW and AVE datasets. In Table 2, we report the direction prediction accuracy of ASMP on the same set of competing methods for both 10 and 28 class setting. Our results clearly attest to the benefit of using the 3D structure of the scene and the motion direction prediction for audio separation. Specifically, ASMP itself (without motion prediction) yields a boost of upto 0.7 dB SDR over prior methods, just on source separation. Additionally, the performance of ASMP improves by atleast another 0.6 dB (on the SDR scale) across the two datasets, when trained to predict the motion direction (Table 1). Furthermore, the results evince that ASMP outperforms prior approaches in almost all of the evaluation categories, by upto 2.0 dB on the SIR scale on the AVE dataset over the next best (and closely-related) approach of AVSGS [8]. These results clearly demonstrate the generalizability of our method to varied datasets, scene contexts, and audio diversity. From amongst the competing techniques, we find that AVSGS comes closest to matching our performance across the two datasets, which is perhaps unsurprising given that they use a 2D scene graph to encode the visual context of the sounding object. However, our results show that the presence of 3D scene geometry is also essential for good audio separation (Table 1 last two rows).

In Table 2, we also report the accuracy of motion prediction of the sounding object (in %) for both ASMP and other baselines for both 10-class and 28-class settings by passing their separated spectrograms through the direction prediction module. As is clear from the table, we find that the accuracy of direction prediction using ASMP is much better than prior methods (by upto $\sim 3\%$). The gains over a simple majority label prediction is especially pronounced, attesting to the efficacy of our direction prediction pipeline.

**Ablation Study:** We report performances of a few key ablated variants of our model in Table 3 on the ASIW dataset, viz. : (i) by computing the weights of the graph edges via a multiscale RBF kernel, (ii) using only the motion direction prediction as a training signal. For the former, we threshold the Chamfer distances at two scales, viz : (a) the median of the weights and (b) $100^{th}$ percentile of the weights. This creates a forest of graphs, which is used in the audio separator network. Our results in Table 3 show that this approach of graph construction levels up to our RBF formulation but falls a bit short, perhaps because of the hard thresholding of edges. In another ablation study, we train only the motion direction prediction loss (i.e., $\lambda_1 = \lambda_2 = \lambda_3 = 0$). As the results in Table 3 show, this variant underperforms our full model, corroborating the utility of the other loss terms.

**Qualitative Results:** In Figure 3, we present sample separation results from the ASIW and AVE test sets, juxtaposing the results obtained by our method against the AVSGS baseline. The superiority of our model's separability is evident even at a glance of the separated spectrograms that mimic the ground truth spectrograms more closely compared to AVSGS. The estimates of the motion directions seem to agree with the ground truth as well. Additional details and results are in the appendix.

## 5 Conclusions

In this work, we presented ASMP, a novel algorithm to leverage visual scene geometry to induce better separation of a mixed audio signal into its semantic constituents. Towards this end, we incorporated pseudo-3D cues into our algorithmic audio separation pipeline via a scene graph data structure. Our work further derives self-supervision cues from the visual dynamics inherent in the videos and use such cues to predict the direction of motion of the sound source from the separated audio; our experiments demonstrate this self-supervision to improve source separation. ASMP demonstrates state-of-the-art performances on two challenging "in-the-wild" datasets. The success at the task of predicting the direction of motion from the visually-guided separated audio suggests ASMP's potential to be used in a variety of other audio-visual problems, including occlusion reasoning, and video generation from audio [7]. We will be making our PyTorch implementation public.

**Limitations:** While ASMP presents promising results on both separation and direction prediction tasks, some caveats remain. For example, our model is not yet capable of effectively inducing source separation for sounds emanating from the background. Further, we require trained object detectors for sound producing classes, which for some rare/uncommon classes maybe hard to obtain.

**Acknowledgements.** MC and NA would like to thank the support of the Office of Naval Research under grant N00014- 20-1-2444, and USDA National Institute of Food and Agriculture under grant 2020-67021-32799/1024178.

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
