# OpenReview forum: "Learning Audio-Visual Dynamics Using Scene Graphs for Audio Source Separation"
_NeurIPS.cc/2022/Conference — NeurIPS 2022 Accept_

### Official Review · Reviewer_XeDg · 2022-07-10

**Rating:** 6
**Confidence:** 4
**Soundness:** 3 good
**Presentation:** 3 good
**Contribution:** 3 good

**Summary:**

This paper proposes a new framework for audio-visual source separation by simultaneously predicting the 3D visual motion of a sounding source. A new Audio Separator and Motion Predictor (ASMP) framework is proposed to leverage the 3D structure of the scene and the
9 motion of sound sources for better audio source separation. Experiments on two challenging audio-visual datasets demonstrate the usefulness of the proposed method.

**Questions:**

- As shown in the 3D scene graph, there are many silent objects like trees, sky, etc. These objects do not make sounds. How does the proposed framework handle silent objects? What is the percentage of silent objects and how will they affect the performance of separation?

- When predicting the direction or 3D moving dynamics of the sounding objects, is the network focusing on the volume of the sound? Is there evidence that the model also makes use of the spatial cues in audio?

**Limitations:**

I cannot find a section to discuss limitations or societal impact.

**Strengths And Weaknesses:**

Strengths:

- The method is well motivated. Jointly predicting the 3D location in space of the sounding source can help to better separate the sounds they make.

- The paper is generally well written with clear and sufficient details.

- Extensive experiments are conducted are two datasets and the proposed method compares favorably to prior methods.

Weakness:

- Title is not the most accurate. Although the key idea is to predict 3D dynamics of sounding objects, but this paper is still more focused on audio-visual source separation. Predicting 3D dynamics is to help the separation learning, so it would be better if the title can indicate this aspect.

- Related work, "Audio Separation Using Direction of Arrival" can be merged into the approach since only two references are discussed there. For better presentation, it can be briefly discussed when presenting the direction prediction network.

- The performance of the proposed method is pretty close to AVSGS as shown in Table 1. AVSGS is also based on scene graphs but with 2D visual context. More analysis or discussions to compare with AVSGS would be desirable.

---

> ### Author Response · Authors · 2022-08-02
> **Response to Reviewer XeDg**
>
> We value the reviewer's encouraging feedback immensely. Below we attempt to address his/her concerns.
>
> **Q1. Title is not the most accurate.**
>
> **A1:** Thanks for your suggestion. Note that the thrust of our work is learning the joint audio-visual dynamics using graphs, which was the key motivation for our title. However, as suggested, we will consider revising it to better reflect our application as well, and update it to “Learning audio-visual dynamics using scene graphs for audio source separation” for example.
>
> **Q2. Related work on "Audio Separation Using Direction of Arrival" can be merged into the approach.**
>
> **A2:** Thanks for the suggestion. We will merge it.
>
> **Q3. More analysis or discussion to compare with AVSGS.**
>
> **A3:**  Besides the difference in model performance between our method (ASMP) and AVSGS [7] as shown in the table 1 of the paper. The following are some key distinctions between the two methods:
> (i) The graphs used in AVSGS are essentially agnostic to scene geometry whereas in ASMP, the edge weights of the graph denote the spatial proximity of the objects (in pseudo-3D) represented by nodes at the end points of an edge. This is key to enabling a sense of 3D direction into the reasoning/separation pipeline. Further, it also allows for injecting sparsity in the graph by discarding the edges with low weights, as shown in Fig. 1 of the supplementary pdf.
> (ii) Moreover, the pseudo-3D grounding of every object in the graph allows us to easily estimate its displacement of sounding objects in that space. This further can be used to derive a training signal, an important feature missing in AVSGS.
> We will include these clarifications in the final paper.
>
>  **Q4. How does the proposed framework handle silent objects? How do they affect the performance of separation?**
>
> | Method | Context Node #1 | Context Node #2 | Context Node #3 | Context Node #5 | Context Node #10 | Context Node #20 |
> |---------------------|----|----|----|----|----|----|
> |ASMP w/o direction  (ASMP-ND) |8.66 |8.71 |8.83 |8.94 |9.15 |9.4 |
>
> **A4:**  In our framework, the list of objects that potentially generate sound, called *auditory objects*, is pre-determined for both datasets. We assume these are objects that produce sound atleast in some videos in our datasets. However, not every occurrence of these objects is associated with a sound. To disambiguate these scenarios, we construct a set of *context objects* (based on pseudo-3D spatial proximity) which determine if an *auditory object* is sounding. For instance, a "guitar" can only be sounding if a "man" node is close to it or else not. Trees, sky, etc. are examples of such *context objects* picked up by the algorithm. They themselves do not make sound but interactions of *auditory objects* with them results in sound. Typically, we choose upto 20 *context objects* per *auditory object* as long as their bounding boxes sufficiently overlap with an *auditory object* in pseudo-3d (denoting spatial proximity).
> To study their effect on separation, we evaluated the ASMP model without direction prediction (ASMP-ND) on the ASIW dataset using SDR. The results of this setting are shown in the table above. From the table we see that with increasing number of *context nodes*, the performance of the model increases, attesting to their importance.
>
> **Q5. When predicting the direction or 3D moving dynamics of the sounding objects, is the network focusing on the volume of the sound? Is there evidence that the model also makes use of the spatial cues in audio?**
>
> **A5:** We believe this sense of motion in sound is attributable to the following: (i) the audio spectrograms that we use are derived from STFT encodings and have the time varying frequency profile in them arising from the object motion, resulting in variations in the amplitude and frequency of the sound source as it approaches or moves away from the microphone (as manifested by, e.g., the Doppler effect), and (ii) the implicit motion cues captured within the scene sub-graph embeddings of sound-producing moving objects in the scene. These embeddings when used to condition the decoder module of the Audio Separation Network, may modulate the generated audio spectrograms using the visual motion directions.
>
> To test the veracity of this hypothesis, we vertically reflected the frames of videos with classes which have a strong directional prior, i.e. objects whose direction of motion exhibit less stochasticity. The flipped classes were: car, train, horse. We report direction prediction performance for all classes as well as these classes (dubbed **Good Class**) without (row 1) and with flipping (row 2) in the table below. We observe only a minor drop in the flipped videos setting validating our insight that the separated audio does incorporate object motion information.
>
> |Row | Method | Dir. Acc. (%) | Good Class Dir Acc. (%) |
> |-|------------------------|----|----|
> |1 |ASMP  |54.6 |66.7 |
> |2 |ASMP w/ flip |52.5 |62.5 |

---

> > ### Comment · Reviewer_XeDg · 2022-08-08
> > **Thanks for the rebuttal**
> >
> > Thanks for the responses. The rebuttal and additional results generally addressed my concerns. I keep my original score and it would be great to include these additional results and analysis in the final version of the paper.

---

> > > ### Author Response · Authors · 2022-08-09
> > > **Grateful for the encouraging response**
> > >
> > > We really appreciate the encouraging response from the reviewer!

---

### Official Review · Reviewer_Zmj2 · 2022-07-10

**Rating:** 7
**Confidence:** 3
**Soundness:** 2 fair
**Presentation:** 4 excellent
**Contribution:** 2 fair

**Summary:**

The paper presents a method for visually-guided audio source separation and motion prediction (exclusively from sound). The method is trained using audio-visual data and operates by constructing a 2.5D scene graph, which is segmented into sub-graphs. Each sub-graph is associated with an audio source. The sub-graph embeddings are subsequently fed to an audio separator network that has a U-Net architecture and produces the separated audio for a particular object. An additional head (direction prediction network) is appended to the audio separator network to predict the motion of the object based on the 2.5D. This allows training using an additional direction prediction loss.

**Questions:**

Since two independent object detectors are used. How is the case where an object is picked up by both detectors (i.e., overlapping bounding boxes) handled?

How is the list of possible auditory classes obtained?

Considering that a mirrored image is equally likely to be associated with the same sound how does the network achieve an accuracy of >50% given that 2 (or more) octants are equiprobable? Also, why does this ambiguity not affect the training of the source separation network? Finally, since the motion prediction network works purely based on sound have the authors tested it on audio that has not been separated but contains only sound from one moving object?

In general, I believe that the method is clearly explained, novel and has been sufficiently evaluated in the paper. However, I do not see how the motion prediction from only mono audio can disambiguate the direction of movement (i.e. how can it tell left from right). I imagine that this ambiguity should also affect the training of the source separation. I would like to hear the authors' response on this before recommending the paper for acceptance.

**Limitations:**

I do not see any serious potential negative societal impact that needs discussing in this work.

**Strengths And Weaknesses:**

## Strengths:

- The paper is well written.
- The use of 3D scene dynamics for audio separation is a novel approach as is motion prediction from audio.
- The evaluation on visually-guided source separation is reasonable. There are ample ablation studies performed in both the main paper and supplementary material. For audio source separation, the method is also tested on YouTube videos and examples are presented in the supplementary material.
- The framework and models used are described in sufficient detail in the supplementary material and a code stub is provided.

## Weaknesses:
- Motion prediction from audio is interesting but the motivation for it is not clear, especially given that the audio is not stereo. This capability of the model I believe has not been sufficiently explored/explained in the paper and raises some questions (see Questions section).

---

> ### Author Response · Authors · 2022-08-02
> **Response to Reviewer Zmj2**
>
> We sincerely value the reviewer's feedback. Please see our responses below addressing all the questions/concerns.
>
> **Q1. Since two independent object detectors are used. How is the case where an object is picked up by both detectors (i.e., overlapping bounding boxes) handled?**
>
> **A1:** In our experiments, this scenario does not arise since the two object detectors are trained to detect complementary, non-overlapping classes. The Open Images detector is trained to only detect musical instruments while the Visual Genome detector detects all other objects. But hypothetically, even if both detectors were detecting overlapping classes, we could use the confidence scores predicted by the detectors to choose the most confident box and use that.
>
> **Q2. How is the list of possible auditory classes obtained?**
>
> **A2:** For our experiments on ASIW, we parse the captions accompanying the videos to extract the list of potential auditory classes following the procedure outlined in [7]. While for AVE, every class label of the original dataset [43] is treated as an auditory class.
>
> **Q2a. Considering that a mirrored image is equally likely to be associated with the same sound how does the network achieve an accuracy of >50% given that 2 (or more) octants are equiprobable? Also, why does this ambiguity not affect the training of the source separation network?**
>
> **A2a:** Thanks for this very insightful question. We believe this is because of the guidance provided by the visual scene graph. Specifically, note that even though we use only a single video frame to construct the scene graph, there are strong directional motion cues present between the sounding object and its context. For example, the position of a train in the image, its appearance, and the direction of tracks suggests the direction of train motion. These implicit motion cues are encoded by the scene-graph embeddings, which are then used by the decoder module of the audio-separator network to modulate the generated spectrogram mask, such that the masked spectrogram incorporates directional cues capturing the object motions. Such cues can be picked up by the direction prediction module (which is trained end-to-end). To test the veracity of this hypothesis, we vertically reflected the frames of videos with classes that have a clear motion direction. The flipped classes were: car, train, horse. We report direction prediction performance for all classes as well as these classes (dubbed **Good Class**) without (row 1) and with flipping (row 2) in the table below. We observe that our model is largely able to adjust to this modified dataset by leveraging appropriate visual cues and registers only a slight drop in the direction prediction performance.
>
> **Q2b. Have the authors tested the motion prediction network on audio that has not been separated but contains only sound from one moving object?**
>
> **A2b:** Yes, we have tested the efficacy of the direction prediction network by training it with the original audio track accompanying single source objects. The results for this experiment are shown in row 3 in the table below. We see that we obtain slightly improved direction prediction performance under this setting.
>
> **Q3. What is the core motivation behind being able to predict direction from mono-audio?**
>
> **A3:** We believe this sense of motion in sound is attributable to an interplay between several cues, namely: (i) the audio spectrograms that we use, are derived from STFT encodings and have the time varying frequency profile in them arising from the object motion, resulting in variations in the amplitude and frequency of the sound source as it approaches or moves away from the microphone (as manifested by, e.g., the Doppler effect), and (ii) the implicit motion cues captured within the scene sub-graph embeddings of sound-producing moving objects in the scene. These embeddings when used to condition the decoder module of the Audio Separation Network, may modulate the generated audio separation masks using the visual motion directions. As a result, the separated audio using these masks could be attuned with directional cues, which are picked up by the direction prediction module. Our results in the flipped videos experiment above, show that the model registers only a slight drop in performance post flipping, suggesting that the synthesized spectrograms are being tuned according to the motion cues gleaned from the scene-graph conditioning feature.
>
> |Row | Method | Dir. Acc. (%) | Good Class Dir Acc. (%) |
> |-|-------------------------------|----|----|
> |1 |ASMP  |54.6 |66.7 |
> |2 |ASMP w/ flip |52.5 |62.5 |
> |3 |Ground truth Audio |56.9 |69.3 |

---

> > ### Comment · Reviewer_Zmj2 · 2022-08-07
> > **Questions regarding direction**
> >
> > I am still concerned about the motion prediction accuracy being above 50%. If indeed this is because visual cues from the videos are encoded in the masked spectrograms then the claim that motion prediction is possible from audio-only can not be made since visual information is used implicitly. Furthermore, I am still puzzled as to why the direction prediction works better than 50% when feeding in the original audio from a clip with a single moving object (denoted as ground truth in the above table) since in this case the separator network and visual information are not used. Finally, if visual information is being used even implicitly through the scene graph then some of the claims and phrases in the paper should be revised since at the moment they lead the reader to believe that motion prediction with >50% accuracy is possible from mono audio alone.

---

> > > ### Author Response · Authors · 2022-08-09
> > > **Follow-up response to Reviewer Zmj2**
> > >
> > > We are very grateful to the reviewer for providing further feedback. Please see our responses below, addressing all the follow-up questions/concerns.
> > >
> > > **Q1. I am still concerned about the motion prediction accuracy being above 50%. If indeed this is because visual cues from the videos are encoded in the masked spectrograms, then can the claim that motion prediction is possible from audio-only be made?**
> > >
> > > **A1:** This is indeed a very insightful observation. In our formulation, different from the vanilla audio-source separation task (such as the cocktail party problem), we leverage visual cues to induce the separation of mixed audio. As a result, when the masked spectrogram (i.e. the output of the Audio Separation Network (ASN) ) is used to predict the direction of motion of the object, indeed that prediction is informed by the implicit motion cues introduced by the visual scene-graph conditioning. We will therefore update the paper to reflect this observation. For instance we will update the abstract to state that: "....In this paper, we propose to use this connection between audio and visual dynamics for solving two challenging tasks simultaneously, namely: (i) separating audio sources from a mixture using visual cues, and (ii) predicting the 3D visual motion of a sounding source using its **separated audio and visual context.**"
> > >
> > > **Q2. Why does the direction prediction work better than 50% when feeding in the original audio from a clip with a single moving object?**
> > >
> > > **A2:** We apologize for the lack of clarity in our response to the precursor to this question above, i.e. *Q2b (Have the authors tested the motion prediction network on audio that has not been separated but contains only sound from one moving object?)*. In the *Ground truth Audio* experiment reported in our response to Q2b, we fed the original audio clip as input to the ASN, rather than feeding it a mixed audio, as is the norm for our setting. Thus, in this scenario, while the ASN is not effectively doing any separation, nonetheless it still modulates the output spectrogram using the visual cues. To study the effectiveness of the direction prediction module in isolation, we conduct a new experiment, where we now feed the original single source audio clip directly as input to the direction prediction network (instead of ASN), to predict the motion direction. The performance of this setting is shown in row 4 in the table below. As alluded to by the reviewer, we observe that this results in a precipitous drop in performance compared to the settings where the output spectrogram is modulated by visual cues. This further confirms our observation that the ASN is indeed injecting valuable signals for direction prediction, derived from the scene-graph into its output spectrogram.
> > >
> > > |Row | Method | Dir. Acc. (%) | Good Class Dir Acc. (%) |
> > > |-|-------------------------------|----|----|
> > > |1 |ASMP  |54.6 |66.7 |
> > > |2 |ASMP w/ flip |52.5 |62.5 |
> > > |3 |Ground truth Audio |56.9 |69.3 |
> > > |4 |Ground truth Audio, no ASN |11.1 |25.0 |
> > >
> > > **Q3. If visual information is being used even implicitly through the scene graph then some of the claims and phrases in the paper should be revised.**
> > >
> > > **A3:** We agree with this suggestion. We will update the paper, to reflect this observation. For instance, statements such as: *"In this paper, we propose to leverage this synergy between sight and sound for solving two challenging tasks simultaneously, viz.: (i) separating audio sources from a mixture using visual cues, and (ii) the novel task of predicting the 3D visual motion of the sounding source using only its separated audio."* (Line 27-30) will be modified to *"In this paper, we propose to leverage this synergy between sight and sound for solving two challenging tasks simultaneously, viz.: (i) separating audio sources from a mixture using visual cues, and (ii) the task of predicting the 3D visual motion of the sounding source using its separated audio and visual context derived from a single frame"*.

---

> > > > ### Comment · Reviewer_Zmj2 · 2022-08-09
> > > > **Increasing my score**
> > > >
> > > > I would like to thank the authors for providing clarity on this point. This was one of the few concerns I had with the paper, which was otherwise well written and presented a novel method. I am raising my score to accept.

---

> > > > > ### Author Response · Authors · 2022-08-09
> > > > > **Grateful for the encouraging response**
> > > > >
> > > > > We sincerely appreciate the insightful suggestions and the encouraging response from the reviewer!

---

### Official Review · Reviewer_j9bT · 2022-07-11

**Rating:** 6
**Confidence:** 4
**Soundness:** 3 good
**Presentation:** 3 good
**Contribution:** 4 excellent

**Summary:**

This paper proposes an audio-visual separation and 3d motion direction prediction model based on 3D visual scene graphs. For visual conditioning, an object detector is run on video reference frames, and depth is predicted from the video frames, both of which are combined to create a "2.5D" scene graph representation. N_u "auditory objects" are identified, and each object defines a subgraph along with their associated "context" nodes. Then, a recurrent network (a GRU) processes the resulting sub-graphs and predicts N_u+1 orthogonal embeddings (N_u auditory sources, and 1 background source). These embeddings are used to condition a U-Net operating on the input mixture spectrogram, where the U-Net conditions on each embedding to predict a mask for the corresponding source. From the separated spectrogram, a 3d motion direction vector is also predicted.

The model uses weak and self-supervised losses trained with the "mix-and-separate" paradigm (i.e. mixtures of 2 videos), including a classification consistency loss, direction prediction loss, co-separation loss (i.e. consistency of predicted masks with ideal oracle masks of the original two source video), and a loss promoting orthogonality of the conditioned embeddings from the GRU.

The proposed approach is tested on the ASIW and AVE datasets, and is shown to improve performance on standard SDR/SIR/SAR separation metrics, and also achieve good performance in terms of predicting the motion direction vector.

Overall I think this is a nice paper with novel contributions, but I think a few important details need to be made more clear.

**Questions:**

## Questions

Q1. What are the reference signals for SDR/SIR/SAR for ASIW and AVE? It seems that you are using a "mix-and-separate" approach that mixes videos together, and the references are the soundtracks of the two source videos? Or is the evaluation performed on single videos, in which case the SDR/SIR/SAR reference is between the sum of the separated sources and the original video's soundtrack? Also, this raises the following questions for me:

Q1a. How many sources per source video?

Q1b. Do these sources always originate from on-screen objects, and if so, how is this determined?

Q1c. What is the level of background noise in videos?

Q1d. Do all videos contain both on-screen and off-screen sound?

Q2. Have you discovered any object classes that the proposed system does not cover (i.e. not contained in the 1600 Visual Genome classes or the Open Images classes)?

Q3. "For each of these auditory objects p_i, we identify the subset V_{p_i} of M − N objects that are non-auditory and overlap": how do you determine if auditory objects are sounding or not? Does the approach assume that some classes of object are always making sound when on-screen?

Q4. "For every video in our setup, we detect upto [sic] two auditory objects, one background auditory object, and 20 context nodes.": Okay, so it seems N_u is a fixed hyperparameter to choose. But I don't see how the consistency loss in Eq. (2) handles less than N_u+1 sources, because it operates over all permutations. Is N_u not fixed, and chosen by how many "auditory" objects are determined to be present? But that doesn't seem to be the case, according to the code stub in the supplementary. This again raises my question Q3, as to how auditory objects are identified.

Q5. "Direction Prediction Loss is a standard cross-entropy loss, which checks for the accuracy of the predicted octant number {0, 1, . . . , 8}": does 0 indicate no motion, and 1-8 mean the 8 octants?

"The visual feature of the background object is derived from a FRCNN embedding of a random crop of the reference frame Fr.": extracting a random crop seems less optimal than extracting a crop that does not overlap with any of the detected objects. Would this work better, and/or did you try this? What if the crop doesn't have visual content correlated with the background sound?

## Minor comments and typos

1. Nit on style: I generally prefer not to capitalize words in an acronym, e.g. "Audio Separator and Motion Predictor (ASMP)" -> "audio separator and motion predictor (ASMP)"

2. "with the kid,": kind of informal

3. "Audio Separation Using Direction of Arrival" section: perhaps mention some works on audio-only beamforming and multi-microphone processing?

4. "Lucak-Kanade" -> "Lucas-Kanade"

5. "A video can therefore, be" -> "A video can therefore be"

**Limitations:**

Limitations in terms of negative societal impact are sufficiently addressed.

Some limitations of the model itself that I think would be good to discuss further are as follows:
1) How many sources the model can handle? I suppose this is set by N_u? I can imagine a lot of real-world scenes, like a concert or busy street, where there are certainly more than 2 sounding objects.
2) Can the model handle videos that only contain background sounds, i.e. contain only off-screen sounds?
3) The model seems to rely on source class to do its separations. Are there source classes that the model does not work on because they are not covered, and can these perhaps be mitigated?

**Strengths And Weaknesses:**

## Strengths

S1. The demo videos are convincing, both in terms of separation and predicting 3d motion direction for sources. I think the 3d motion direction prediction is particularly impressive.

S2. The paper is clear and well-written.

S3. The results on ASIW and AVE clearly show improvement in terms of SDR/SIR/SAR.

S4. Paper performs ablations and a user study.

## Weaknesses

W1. The system is quite complicated, with a lot of components. However, they are generally described well, and the approach is effective according to the objective results and demo videos.

W2. More detail about the ASIW and AVE datasets that are relevant to the proposed separation task are required. See my Q1 below. I would be inclined to increase my score if these were made more clear.

W3. Some details about the approach are still a bit unclear. Please see my questions below. I would be inclined to increase my score if these were made more clear.

W4. It would be good to discuss more limitations of the method. See Limitations section below.

---

> ### Author Response · Authors · 2022-08-02
> **Response to Reviewer j9bT**
>
> We appreciate the reviewer's insightful questions and valuable feedback. Below we address all the concerns and questions.
>
> **Q1.  What are the reference signals for SDR/SIR/SAR?**
>
> **A1:** For quantitative evaluation, we follow standard evaluation protocols and test competing models, using the "mix-and-separate" framework, on their ability to separate the sound tracks of two single-source videos. In such a setting, the audio input to the algorithms is the average of the two audio tracks while the original audio-tracks of these videos become the ground-truth.
>
> **Q1a.  How many sources per source video?**
>
> **A1a:** For the ASIW dataset, we assume that every video has at most 2 auditory sources and 1 background while for the AVE dataset, we assume 1 source and 1 background.
>
> **Q1b. Do these sources always originate from on-screen objects? How is this determined?**
>
> **A1b:** Yes, the auditory sources are always on-screen. To determine the list of these objects, for ASIW, we leverage the captions accompanying these videos and follow the post-processing procedure listed in [7]. For AVE, we use the class label associated with every video [43] as its *auditory object*.
>
> **Q1c. What is the level of background noise in videos?**
>
> **A1c:** This really varies across videos. Some of these, such as the musical instrument videos in AVE, have little to no background sound, while some others, like videos of cars honking on the street in ASIW, have significant background sounds.
>
> **Q1d. Do all videos contain both on-screen and off-screen sound?**
>
> **A1d:** On-screen sound is present in all videos that we conduct experiments on. However, off-screen sounds may or may not be present.
>
> **Q2. Have you discovered any object classes that the system does not cover?**
>
> **A2:** The set of *auditory objects* detected by our two detectors cover the principal auditory sources of all videos across the ASIW and AVE datasets. However, rare auditory objects such as dolphins, are currently not covered by our detectors.
>
> **Q3. How do you determine if auditory objects are sounding or not? Do some classes of object always make sound?**
>
> **A3:** At training time, the ground-truth of which *auditory objects* are sounding, is determined from the video captions for ASIW or class labels for AVE. However, for prediction, the model uses pseudo-3D spatial context (i.e. Chamfer based edge weights), to automatically determine the sounding objects. For instance, if a "guitar" node is very far away from "man" node, then the model determines the guitar as not to be sounding. Thus *auditory objects* do not become sounding by default, for purposes of prediction.
>
> **Q4. How does Eq. (2) handle less than N_u+1 source? Is N_u not fixed?**
>
> **A4:** At training time, we run the forward pass of the RNN to extract ($N_u  + 1$) embeddings for each video. However, if a particular video has $N_p$ sources, with $N_p < N_u$ (as determined from the video metadata like captions or labels), then the permutation is over the first $N_p + 1$ predictions of the model. In such a setting, the loss over the excess $N_u - N_p$ terms is masked.
>
> Yes, $N_u$ is fixed.
>
> **Q5. Does 0 indicate no motion, and 1-8 mean the 8 octants?**
>
> **A5:** Yes.
>
> **Q5. How does the model perform with non-overlapping background bounding boxes?**
>
> **A5:** Thanks for this insightful question. To study this, we conducted experiments on ASIW where the background object features were derived from a fixed-sized bounding box from an image in an unrelated dataset  (ADE20k). We used the feature embedding of this box to represent the background class node in all our scene graphs, thus ensuring there is no bounding box overlap. Row 2 in the table below, presents the results. We see that the model exhibits a very minor boost in performance, suggesting that while this change is not a critical shortcoming, nonetheless it potentially helps the model learn a less ambiguous mapping between object features and their audio.
>
> |Row | Method | SDR | SIR | SAR |
> |-|-------------------------------|----|----|----|
> |1 |ASMP (full)|9.9 |15.3 |14.1 |
> |2 |ASMP non-overlap bkg |10.1 |15.4 |14.3 |
> |3 |ASMP w/ N_u=3 |9.5 |14.7 |13.3 |
>
> Limitations:
>
> **L1. How many sources the model can handle?**
>
> **A1:**  $N_u$ determines this. For ASIW, $N_u = 2$, while for AVE it is 1. The results for $N_u=3$ for ASIW are reported in row 3 of the table above. As we see, this does not result in too much of a performance change perhaps because the dataset was built mostly with videos with 2 sound sources [7].
>
> **L2. Can the model handle videos that only contain background sounds?**
>
> **A2:**  Yes, it can. Although, in such settings the sub-graphs are trivially the full graphs.
>
> **L3. Are there source classes that the model does not work on? How to mitigate this?**
>
> **A3:**  Yes, these would be classes not covered by the detector. E.g. dolphin. Currently, to mitigate this weakness one has to update the detectors to be able to detect such classes.

---

> > ### Comment · Reviewer_j9bT · 2022-08-08
> > **Response to authors, and a couple comments**
> >
> > Hi there, thanks for the detailed responses, they helped clear up my questions. Thanks too for the additional results with non-overlapping background bounding boxes.
> >
> > A few additional comments:
> >
> > > A1: For quantitative evaluation, we follow standard evaluation protocols and test competing models, using the "mix-and-separate" framework, on their ability to separate the sound tracks of two single-source videos. In such a setting, the audio input to the algorithms is the average of the two audio tracks while the original audio-tracks of these videos become the ground-truth.
> >
> > Taking the mean of the two audio soundtracks, and using the two original audio soundtracks as references, introduces a scale error between the sources in the audio mixture and the reference sources, and I would think a sum would be better. But this probably won't affect results, because SDR/SIR/SAR are used. These metrics can account for gain errors, since they allow filtering of the reference sources with a 512-tap FIR filter.
> >
> > I would also encourage the authors to include the information provided in response to my Q1 in the paper.
> >
> > > A3: At training time, the ground-truth of which auditory objects are sounding, is determined from the video captions for ASIW or class labels for AVE. However, for prediction, the model uses pseudo-3D spatial context (i.e. Chamfer based edge weights), to automatically determine the sounding objects. For instance, if a "guitar" node is very far away from "man" node, then the model determines the guitar as not to be sounding. Thus auditory objects do not become sounding by default, for purposes of prediction.
> >
> > Wow, that sounds potentially complicated to enumerate all the possible interactions that indicate sounding objects, and potentially brittle to interactions not modeled. I think it would be good to have these rules for sounding objects enumerated somewhere, perhaps in the code or supplementary.
> >
> > > L2. Can the model handle videos that only contain background sounds?
> > > A2: Yes, it can. Although, in such settings the sub-graphs are trivially the full graphs.
> >
> > Perhaps you could consider an "off-screen-only" set of evaluation examples, to measure the model's ability to reject background sound, and to not make "false alarm" errors when no sound objects are actually present?

---

> > > ### Comment · Reviewer_j9bT · 2022-08-09
> > > **Also raising score**
> > >
> > > Also, given that the authors' response cleared up my questions, I am raising my score a bit. Thanks again for the responses.

---

> > > > ### Author Response · Authors · 2022-08-09
> > > > **Grateful for the encouraging response**
> > > >
> > > > We really appreciate the heartening feedback from the reviewer!

---

> > > ### Author Response · Authors · 2022-08-09
> > > **Follow-up response to Reviewer j9bT**
> > >
> > > We are sincerely grateful to the reviewer for providing further feedback. Please see our responses below as follow-up to the new suggestions.
> > >
> > > **Q1. Taking the mean of the two audio soundtracks, and using the two original audio soundtracks as references, introduces a scale error between the sources in the audio mixture and the reference sources, would a sum be better?**
> > >
> > > **A1:** Indeed, the mean audio signal does introduce the potential of a scale error. To adjudicate if taking a sum of the input audio tracks results in better separation, we re-evaluate the ASMP model with non-overlapping background (ASMP non-overlap bkg), reported above, providing the sum of the two audio samples as input to the separation network, instead of their mean.  We denote this model by *ASMP non-overlap bkg, no avg* and report its performance in row 3 in the table below. From the table, we observe that switching the input audio from mean to the sum of the audio of the constituent signals results in a negligible change in audio source separation quality, atleast as measured by SDR/SIR/SAR. Moreover, part of the change in performance is attributable to the train/test mismatch as well, since the model was trained with the mean audio signal as input but tested with the sum. These analyses would be included in the final version of the paper.
> > >
> > > |Row | Method | SDR | SIR | SAR |
> > > |-|-------------------------------|----|----|----|
> > > |1 |ASMP (full)|9.9 |15.3 |14.1 |
> > > |2 |ASMP non-overlap bkg |10.1 |15.4 |14.3 |
> > > |3 |ASMP non-overlap bkg, no avg |10.0 |15.6 |13.9 |
> > >
> > > **Q2. Can the rules governing the definition for sounding objects be enumerated somewhere?**
> > >
> > > **A2:** We will update the supplementary pdf to include the following lines, enunciating the process of identifying training time sounding objects: *"At training time, the ground-truth of which auditory objects are sounding, is determined from the metadata of the videos. For ASIW, we parse the captions accompanying the video to extract the names of the objects making sound, following the process laid out in [7] (results in atmost a couple of sources per video) while for the AVE dataset, the sounding object is trivially derived from the class label associated with a video."*
> > >
> > > The following lines would describe the sounding object determination process for prediction, which leverages the graph sparsity introduced by our formulation to speed up the process: *"For prediction, our model uses pseudo-3D spatial context, as derived from the Chamfer distance based edge weights, to automatically determine which objects are sounding and which are not. For instance, if a "guitar" node is very far away from "man" node, then the model determines the guitar as not to be sounding. Since edges with low weights are indicative of large physical separation, we use this as a prior to discard such edges from our original fully-connected graph (covering all interactions), which in turn speeds up the process of determining sounding objects. We found this sparsity prior successful in capturing all the critical interactions and thereby preserving performance, as shown in Fig. 1 of the supplementary pdf."*
> > >
> > > **Q3. Perhaps you could consider an "off-screen-only" set of evaluation examples, to measure the model's ability to reject background sound?**
> > >
> > > **A3:** This is a great suggestion to demonstrate the efficacy of our method. We will update the paper to include results for "off-screen only" evaluation on a few videos captured in the wild (collected from the web) where we expect such sounds to not be separated into its constituents but be treated as background audio in its entirety.

---

### Official Review · Reviewer_bcMT · 2022-07-16

**Rating:** 5
**Confidence:** 3
**Soundness:** 3 good
**Presentation:** 3 good
**Contribution:** 2 fair

**Summary:**

This paper proposes to model the dynamics of audio-visual objects in the scene with a pseudo-3D scene graph. The model uses a RNN to roll out multiple features to represent the audio-visual features of different objects. This paper also estimates the ground truth sound source motion from the video and adds a direction prediction loss for predicting the direciton of motion from the separated sound source. Authors show that the proposed model outperfroms SOTA models on two in the wild audio-visual separation datasets.

**Questions:**

Why the features rolled out RNN from one single global feature is better than more object centric features?

Why is it posible to predict direction from single-channel audio?

Can we see how the baselines do compared to your model for the same qualitative videos?

**Ethics Review Area:**

["I don’t know"]

**Limitations:**

Yes, both are discussed.

**Strengths And Weaknesses:**

The idea of constructing a scene graph for audio-visual separation is interesting and does go beyond the traditional object based audio-visual separation pipelines. The paper provides sufficient details for modeling and implementation. The performance improvement over the baseline also seems pretty impressive.

However, I have doubts about the two proposed contributions. First of all, constructing scene graphs and learning audio-visual dynamics for separating sounds make sense. However, whether use RNN to generate multiple features from one global feature to represent differnet objects is questionable because the rollout is not conditioned on anything else. How do you make sure the generated features correspond to visual objects? It's not obvious to me why this representation is much better than object-centric features with some local motion features.

Secondly, why is possible to predict direction from single-channel audio? The best scenario is you can predict whether the sound is moving closer to or away from the microphone. Even for that case, you need to assume the volume for the source sound is constant, which is certainly not true in many cases. I don't think it's possible to predict directions just from single-channel audio and if it does, then it's really questionable why it does.

For the qualitative video, it would also be better to display how your model does compared to the baselines. The separation results do not sound great to me, but it could be because of the lack of comparisons.

Another issue is also that the paper has a very complicated pipeline, which makes it hard to verify it's the proposed components that led to the performance improvement.

---

> ### Author Response · Authors · 2022-08-02
> **Response to Reviewer bcMT**
>
> We are sincerely grateful to the reviewer for the insightful suggestions. Please find below our responses to all the questions/weaknesses flagged.
>
> **Q1a. How does the RNN generate multiple features from one single global feature?**
>
> **A1a:** Our insight into the design of this RNN module is inspired by the classic orthogonal matching pursuit (OMP) method. Analogous to OMP, we desire our graph encoding operations to embed the graph into well-separated sub-graph embeddings in a latent-space. The RNN (which is trained with the orthogonality loss in Eq. 1) is tasked to pick one latent graph embedding in each of its roll out step. The hidden state of the RNN allows keeping track of the number of steps for which the RNN has been rolled out; thereby producing a latent embedding for a different scene sub-graph at each step.
>
> **Q1b. How do you make sure that the generated RNN features correspond to visual objects?**
>
> **A1b:** Recall that at training time, the RNN embeddings are encouraged to be mutually orthogonal while ensuring that each such embedding results in a separated audio source via the consistency loss (Eq. 2 in the paper). Moreover, the cyclic loss ensures that the separated audios when mixed back yield the same mixed audio as the input to the Audio Separator Network (ASN) network. The only way all these constraints are simultaneously satisfied is if the embeddings at successive steps encode the visual representation of the different sound sources.
>
> **Q1c. Why are the RNN features, derived from a single global feature, better than more object-centric features?**
>
> **A1c:** The advantage of our visual feature encoding scheme, in contrast to object-centric features (such as [11]) is in capturing *interaction sounds*. These are sounds resulting from inter-object interactions, such as a drummer and his/her drum kit, in contrast to single object sounds such as dog barking. Our approach captures these interactions via graph edges while the RNN produces feature embeddings that dynamically adjust the number of *context objects* for every potential sound source, as per the data. Object-centric input features would struggle to show such adaptability (even if a fixed number of context object features are hard coded into them).
>
> We quantitatively compare the two settings on the ASIW dataset, when both schemes are tasked to also predict the object motion in rows 1 and 7 in the table below. The results show that the object-centric features [11] to be inferior to our method, demonstrating that contextual information improves the separation quality significantly.
>
> |Row | Method | SDR | SIR | SAR | Dir. Acc. (%) |
> |-|-------------------------------|----|----|----|----|
> |1 |ASMP (full)|9.9 |15.3 |14.1 |54.6 |
> |2 |ASMP w/o 3D info, w/o direction  (ASMP-N3D,ND) |8.8 |14.1 |13.0 |- |
> |3 |ASMP -Flip |9.8 |15.1 |13.7 |52.5 |
> |4 |ASMP-N3D,ND - No Skip Conn. in ASN |2.8 |4.6 |11.3 |- |
> |5 |ASMP-N3D,ND - No Scene Graph |6.5 |11.6 |11.8 |- |
> |6 |ASMP-N3D,ND - No GRU| 6.5| 12.3| 10.6|- |
> |7 |Co-Separation [11] |6.6 |12.9 |12.6 |42.9 |
>
> **Q2. Why is it possible to predict direction from single-channel audio?**
>
> **A2:** We believe this sense of motion in sound is attributable to the following: (i) the audio spectrograms that we use are derived from STFT encodings and have the time varying frequency profile in them arising from the object motion, resulting in variations in the amplitude and frequency of the sound source as it approaches or moves away from the microphone (as manifested by, e.g., the Doppler effect), and (ii) the implicit motion cues captured within the scene sub-graph embeddings of sound-producing moving objects in the scene. These embeddings when used to condition the ASN module, may modulate the generated audio spectrograms using the visual motion directions.  To ascertain the latter insight, we flipped the video frames vertically in a few sound classes (car, train, and horse) of the ASIW dataset which exhibit less stochasticity in motion direction. The results, in row 3 of the table above, show that the predicted direction is also flipped in most of the cases since the performance drops only slightly, validating our insight.
>
> **Q3. Can we see how the baselines do compared to your model for the same qualitative videos?**
>
> **A3:** We show qualitative comparisons between our method (ASMP) and our closest competitor (AVSGS) [7] in the supplementary video. These results are available from 5 min. 29 sec onwards. The results clearly demonstrate that ASMP is better in audio source separation across both datasets.
>
> **Q4. What components in the model lead to performance gains?**
>
> **A4:** In rows 4 - 6 of the table above, we show model performances with ablated variants of the ASMP's architecture on the ASIW dataset. The results underscore the importance of these modules, with the skip connections in the ASN block being the most critical. We also present ablations on the various loss terms in Table 2 of supplementary pdf.

---

> > ### Comment · Reviewer_bcMT · 2022-08-09
> > **Thanks for the rebuttal**
> >
> > Thank you for the rebuttal. The explanation and ablations numbers partially answered my questions, but I still have some doubts about the directionality of the sound contained in single channel audio. The variation in the amplitude and frequency of the audio does not only come from the motion but also come from the emitted signal (unless the emitted signal remains constant over time). I'm keeping my rating and it would be great if authors could incorporate the feedback into the paper.

---

> > > ### Author Response · Authors · 2022-08-09
> > > **Follow-up response to Reviewer bcMT**
> > >
> > > We are sincerely grateful to the reviewer for acknowledging the points made in our rebuttal response and for providing further feedback. Please see our responses below to the follow-up comments/concerns.
> > >
> > > **Q1. The variation in the amplitude and frequency of the audio does not only come from the motion but also from the emitted sound signal (unless the emitted sound signal remains constant over time). Then how can the direction of motion be predicted from this varying sound source?**
> > >
> > > **A1:** Indeed, we agree with the reviewer's observation that variations in the amplitude and the frequency of the separated spectrogram, emerging from the Audio Separator Network (ASN), may not be arising only from the motion of the audio source but also from changes in the sound signal itself, produced by the source. In our current setup, we assume the audio from a source is stationary at a semantic level (i.e., for example, the sound of a barking dog running towards its human owner may have variations in the amplitude and frequency of the barking sound, but the barking signal still retains enough attributes such that as the owner can still decipher if the dog is coming closer or moving farther away), and the motion cues as provided by the visual graph module can disentangle and modulate the motion features exclusively, separately from any variation in the emitted audio signal. We believe this disentanglement is perhaps not perfect, and that could be a reason our direction prediction accuracy is only 54.6% on the ASIW dataset.  We emphasize that the direction prediction leverages implicit motion cues in the modulated spectrogram to predict the direction of motion. We will update the paper clarifying your insightful and key observations.
> > >
> > > **Q2. It would be great if authors could incorporate the feedback into the paper.**
> > >
> > > **A2:** We deeply value the suggestions by the reviewer and will incorporate them into the paper.

---

### Comment · Area_Chair_noHH · 2022-08-08
**Reviewers bcMT, j9bT, and XeDg - please reply to the authors**

The authors have responded to questions that you raised:
- bcMt raised the question of how the RNN can generate different features given the same global features, how the generated features are forced to correspond to objects, why the representation would be better than object-centric features with local motion, and how direction can be predicted from single-channel audio; requested a qualitative comparison of the proposed model to the baselines; and asked what aspects of the proposed model lead to performance improvement
- j9bT asked what the references were for SDR, SIR, and SAR; asked how many sources there were per video; asked if the auditory sources are always on screen; asked about levels of background noise; asked if all videos include both on- and off-screen sounds; asked if there are objects that are not covered by the proposed system; asked how it is determined if objects are sounding or not; asked how fewer than N_u+ 1 sources can be handled and if N_u is fixed; asked about the meaning of the direction prediction labels; suggested using a nonoverlapping bounding box to extract background visual features; asked how many sources can be handled; asked if videos containing only background sounds can be handled; and asked if unmodeled source classes can be mitigated
- XeDg raised concerns about the accuracy of the paper title, suggested simplifying the structure of the related work, requested more careful comparisons to AVSGS, asked how silent objects are handled, and asked if spatial cues in audio are exploited

As we are near the end of the author-reviewer discussion period, I ask that you please read the authors' reply and respond promptly.

Thanks to reviewer Zmj2 for having already responded to the authors

---

### Meta-Review · Area_Chair_noHH · 2022-08-25

**Recommendation:** Accept
**Confidence:** Certain

**Metareview:**

All four reviewers agree that this paper demonstrates strong improvements over prior methods, and there is broad agreement among the reviewers that the model is well motivated and novel, and that the paper is clearly written.

There was a good discussion between the authors and reviewers on a number of perceived weaknesses in the paper, and the authors were able to address these concerns with additional experiments and proposed revisions, prompting two reviewers to raise their scores. In the end, all reviewers recommend acceptance to some degree, and in my judgement, the most negative reviewer (who recommends borderline accept) has missed the point, made both in the paper and during the discussion, that the estimation of source direction from single-channel audio depends not only on audio cues, but also on video cues.

**Award:**

No

---

### Decision · Program_Chairs · 2022-09-14

Accept